# Neural tuning matches frequency-dependent time differences between the ears

**Victor Benichoux[1,2,3,4]\*, Bertrand Fontaine[5,6], Tom P Franken[6], Shotaro Karino[6,7], Philip X Joris[6]\*[†], Romain Brette[1,2,3,4]\*[†]**

[1]Institut d'Etudes de la Cognition, Ecole Normale Supérieure, Paris, France; [2]Université Pierre-et-Marie-Curie, UMR_S 968, Institut de la Vision, Sorbonne Universités, Paris, France; [3]U968, Institut de la Vision, Institut National de la Santé et de la Recherche Médicale, Paris, France; [4]UMR_7210, École des Neurosciences de Paris Île-de-France, Centre National de la Recherche Scientifique, Paris, France; [5]Dominick P. Purpura Department of Neuroscience, Albert Einstein College of Medicine, New York, United States; [6]Laboratory of Auditory Neurophysiology, University of Leuven, Leuven, Belgium; [7]Department of Otolaryngology, Faculty of Medicine, University of Tokyo, Tokyo, Japan

**Abstract** The time it takes a sound to travel from source to ear differs between the ears and creates an interaural delay. It varies systematically with spatial direction and is generally modeled as a pure time delay, independent of frequency. In acoustical recordings, we found that interaural delay varies with frequency at a fine scale. In physiological recordings of midbrain neurons sensitive to interaural delay, we found that preferred delay also varies with sound frequency. Similar observations reported earlier were not incorporated in a functional framework. We find that the frequency dependence of acoustical and physiological interaural delays are matched in key respects. This suggests that binaural neurons are tuned to acoustical features of ecological environments, rather than to fixed interaural delays. Using recordings from the nerve and brainstem we show that this tuning may emerge from neurons detecting coincidences between input fibers that are mistuned in frequency.

\*For correspondence: victor. benichoux@ucdenver.edu (VB); Philip.Joris@med.kuleuven.be (PXJ); romain.brette@inserm.fr (RB)

[†]These authors contributed equally to this work

**Reviewing editor**: David C Van Essen, Washington University in St Louis, United States

## Introduction

Acoustical waves produced by a sound source reach the two ears at slightly different times depending on its spatial position. Interaural time differences (ITDs) are used by many species to localize sounds in the horizontal plane. In mammals, neurons in the medial superior olive (MSO), just three synapses away from the cochlear receptors, are sensitive to both ITD and sound frequency. It is thought that their activity encodes ITD in a frequency band, and is then interpreted in terms of spatial position. They project to neurons in the inferior colliculus (IC), which inherit these properties.

The firing rate of these neurons is strongly modulated by the ITD of a tone presented binaurally through earphones. For a 600 Hz tone, the neuron shown in *Figure 1A*, recorded in the IC of a cat, responds maximally at a 'best ITD' of 345 μs, close to the maximum natural ITD reported for cat (about 350 μs) (*Roth et al., 1980*). In the Jeffress model, the textbook model of ITD processing (*Jeffress, 1948*; *Joris and Yin, 2007*), ITD tuning arises from the detection of coincidences between spikes relayed from auditory nerve fibers tuned to the same frequency at the two ears, and the

**eLife digest** When you hear a sound, such as someone calling your name, it is often possible to make a good estimate of where that sound came from. If the sound came from the left, it would reach your left ear before your right ear, and vice versa if the sound originated from your right. The time that passes between the sound reaching each ear is known as the 'interaural time difference'. Previous research has suggested that specific neurons in the brain respond to specific interaural time differences, and the brain then uses this interaural time difference to locate the sound.

Sounds come in various frequencies from high-pitched alarms to low bass tones, and how a neuron responds to interaural time differences appears to change according to the frequency of the sound being played. For example, a given neuron may respond to a 200- microsecond interaural time difference when a tone is played at a high frequency, but show no response to this time difference when the tone is played at a low frequency. To date, researchers had been unable to explain why this occurs.

Here, Benichoux et al. investigated this topic by playing a variety of sounds to anaesthetized cats. Electrodes were used to record the responses of individual neurons in the cats' brains, and the properties of the sound waves that reached the cats' ears were also recorded. These experiments revealed that the time it took a sound to travel from a location to each of the cats' ears, and consequently the interaural time difference, depended on whether it was a high-pitched or a low-pitched sound. This happened because different properties of the environment, such as the angle of the cat's head, affected specific frequencies in different ways.

As expected, the neurons' responses were also affected by sound frequency. Indeed, the neurons' behaviour mirrored that of the sound waves themselves. This shows that neurons do not, as previously thought, simply react to specific interaural differences. Instead, these neurons use both sound frequency and interaural time differences to produce a thorough approximation of the sound's location. The precise mechanisms that generate this brain adaptation to the animal's environment remain to be determined.

neuron responds maximally when the sound's ITD equals the mismatch in axonal conduction delay between inputs from the two cochleae. This model predicts that, for a given neuron, ITD tuning is independent of the sound's frequency. However, at 900 Hz, the neuron of *Figure 1* is tuned to an ITD of 158 µs and barely responds to an ITD of 345 µs, while at 400 Hz the neuron responds maximally at 500 µs and is much less responsive at 345 µs (*Figure 1A*). In fact, the range of best ITDs that this neuron shows at different stimulus frequencies spans several hundred µs (*Figure 1B*), which is large considering that the maximum natural ITD in cats is ~350 µs. Thus, the ITD tuning of this neuron varies broadly with sound frequency. This property has been observed in binaural neurons of many species, including cats (*Yin and Kuwada, 1983*), guinea pigs (*McAlpine et al., 1996*; *Palmer and Kuwada, 2005*), rabbits (*Kuwada et al., 1987*), chinchilla (*Bremen and Joris, 2013*), gerbils (*Day and Semple, 2011*) and dogs (*Goldberg and Brown, 1969*), but no functional significance has been associated with it. Readers should note that this property is observed *within* neurons as a function of frequency, and differs from the *population* property that has also been widely observed, where neurons tuned to low frequencies tend to have larger best ITDs than high-frequency neurons (*McAlpine et al., 1996*; *Hancock and Delgutte, 2004*; *Joris et al., 2006*; *Day and Semple, 2011*; *Bremen and Joris, 2013*).

On the other hand, it is known that in natural environments, the acoustic ITD itself varies not only with spatial position but also with frequency, due to sound diffraction by the head (*Roth et al., 1980*) and early reflections from the ground (*Gourévitch and Brette, 2012*). Here we show with acoustical recordings and simulations that the variation of ITD with frequency can be substantial at the scale of a single neuron's receptive field. We then show that the detailed statistics of this variation are matched by the tuning of binaural neurons. Finally, we show that slight mismatches in the frequency tuning of auditory nerve fibers projecting to binaural neurons are a plausible mechanism to explain tuning to complex binaural features of ecological environments, and we show the existence of asymmetries in the spectral properties of MSO inputs using intracellular recordings.

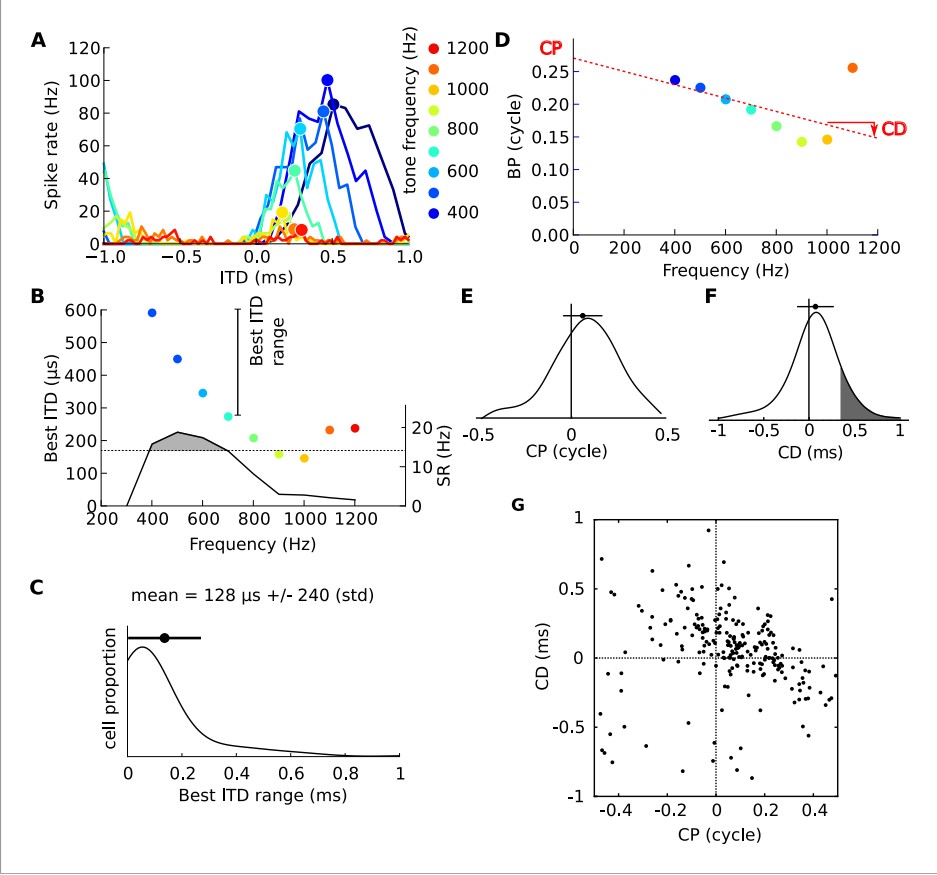

**Figure 1**. Frequency-dependence of best delays. (**A**) Firing rate vs ITD for one neuron, to tones between 400 Hz (blue) and 1200 Hz (orange). (**B**) Best interaural time difference (ITD) (colored dots, left axis), and sync-rate (SR), (black line, right axis) vs frequency for the same cell. Data points with SR higher than 80% of the maximum value are used to calculate the range of best ITD (shaded area above dotted line). (**C**) Distribution of the range of best ITDs across all 186 cells. (**D**) Best phase (BP) vs frequency and linear regression. The characteristic phase (CP, here 0.27 cycle) is the intercept; the characteristic delay (CD, here –0.102 ms) is the slope. (**E**) Distribution of CP across all cells (N = 186). (**F**) Distribution of CD. (**G**) CD vs CP across all cells.

The following figure supplements are available for figure 1:

**Figure supplement 1**. Linearity of BP vs frequency curves.

**Figure supplement 2**. Statistical significance of CP-CD correlation.

**Figure supplement 3**. Best frequency (BF) and characteristic frequency (CF) of recorded cells.

**Figure supplement 4**. BP vs tone frequency for 13 sample cells.

## Results

### Frequency dependence of neural tuning

We examined ITD tuning in 186 IC neurons of cats tuned at a characteristic frequency (CF, frequency of lowest rate threshold) between 100 and 3300 Hz (see *Figure 1—figure supplement 3*). We found that the best ITDs of a neuron, at the different frequencies to which they were sensitive, spanned on average a range of 128 µs (± 240 µs) (*Figure 1C*). This extent is large, considering the maximum natural ITD reported for cats (~350 µs) and their ability to discriminate ITDs differing by only 20 µs (*Wakeford and Robinson, 1974*).

The dependence of best ITD on sound frequency can be analyzed more precisely (*Yin and Kuwada, 1983*). The best ITD can be expressed relative to the period of the tone's frequency f, and is then called the best phase (BP): BP = best ITD × f (*Figure 1D*). For a neuron with a fixed best ITD that does not depend on frequency, BP is a linear function of frequency with 0 y-intercept. But the neuron shown in *Figure 1D* does not fit this simple relationship: a better fit is a linear relationship with an offset at 0 Hz, called the characteristic phase (CP), measured between −0.5 and 0.5 cycle. We computed circular-linear regressions for all 186 neurons, which were highly significant in most cases (*Figure 1—figure supplement 1*; see also other examples on *Figure 1—figure supplement 4*). We found that CP was broadly distributed across an entire cycle (*Figure 1E*), indicating that the best ITD of many neurons is not fixed but depends on frequency. The slopes of linear regressions are called characteristic delays (CD; *Figure 1D,F*) (*Yin and Kuwada, 1983*). If neurons were tuned to fixed ITDs in the contralateral field, we would expect CDs to be distributed between approximately 0 µs and 350 µs. In our neurons, the CDs are mainly positive (corresponding to contralateral leading sounds) and mostly within the natural range of 350 µs, but a minority of cells have negative CDs (38%) and a smaller minority have CDs larger than 350 µs (19%, grey area in *Figure 1F*). Most intriguingly, CDs are negatively correlated with CPs (*Figure 1G*). We checked with bootstrap analysis that this negative correlation is not due to measurement artifacts (*Figure 1—figure supplement 2*). All these observations are consistent with previous findings in other species (*Yin and Kuwada, 1983*; *McAlpine et al., 1996*; *Palmer and Kuwada, 2005*).

## Frequency dependence of ITD

We looked in the acoustics for a functional rationale for the frequency-dependence of neural tuning to ITD within single neurons. It is known that the acoustic ITD itself varies not only with spatial position but also with frequency, due to sound diffraction by head and body (*Roth et al., 1980*). This variation can be quantified by analyzing head-related transfer functions (HRTFs), which measure the acoustical filtering of the head and body for sources at various positions. *Figure 2A* shows the variation of phase ITD with frequency for different source directions in an anaesthetized cat (*Tollin and Koka, 2009*). The 'phase' ITD reported here is the value of the ITD of a pure tone stimulus at a given frequency (see 'Materials and methods' for additional ITD definitions). These patterns are consistent with previous acoustical measurements in cats using tones (*Roth et al., 1980*). We also found similar patterns in high-resolution recordings on a taxidermist model of a cat with a natural posture (*Figure 2B*). We checked that these patterns were not due to possible limitations of acoustical recordings by comparing them with numerical simulations of HRTF obtained on a 3D model of the same cat (*Rébillat et al., 2014*) (*Figure 2C,D*). The global structure of these patterns is consistent with a spherical model of the head (*Kuhn, 1977*) (*Figure 2E*). However, their fine structure depends on posture (*Figure 2C,D*, cat's head in a different position), on the presence of a ground (*Figure 2E,F*, spherical head model without and with a ground plane), and on whether the source is in the front or in the back (*Figure 2A–D*, solid vs dashed curves)—because of reflections on the body of the cat. We remark that reflections on the ground or on the body of the cat come too early to be separated from the direct signal (*Gourévitch and Brette, 2012*) and they must be considered as an integral part of the binaural signal received by the animal. Thus in ecological conditions, the ITD generally varies with frequency for a given source position.

Natural signals do not consist of a single frequency but typically have a certain bandwidth, and individual binaural neurons integrate signal information from a range of frequencies, which matches the bandwidth of filtering in the cochlea (*Mc Laughlin et al., 2007*). In view of the dependence of the acoustical binaural cue (ITD) on frequency (*Figure 2*), the next question becomes how this compares to the neural dependence of best ITD on frequency. We tested whether the features seen in the electrophysiological data could be explained by the hypothesis that neurons are tuned to frequency-dependent ITDs as found in ecological environments. This is illustrated in *Figure 3* which shows frequency-dependent ITDs and zooms in on the frequency band 600–1000 Hz for 3 azimuths. If a neuron were tuned to a fixed ITD (e.g., 325 µs, in *Figure 3B*, top), then it would be most responsive to different azimuths at different frequencies. On the other hand, if the neuron were tuned to a fixed spatial azimuth, then its best ITD would vary with frequency to match the variation of ITD with frequency at that position (*Figure 3B*, bottom). Since the relationship between ITD and frequency is not fixed but depends on variables in listener, source, and environment—as illustrated in *Figure 2*—we looked for statistical correspondences between acoustical and neural measurements.

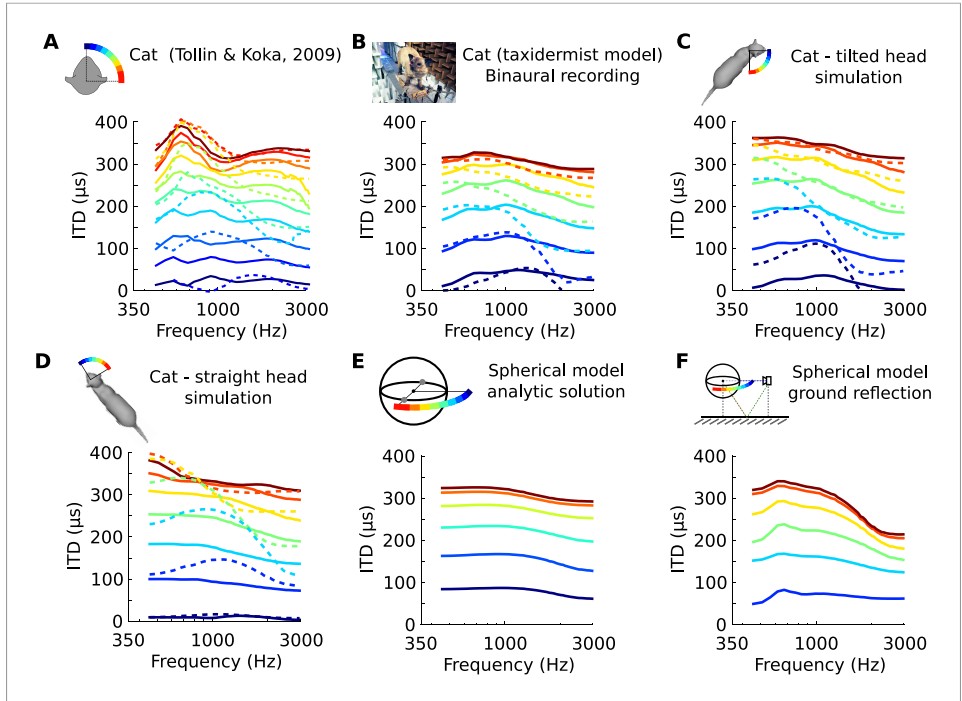

**Figure 2**. Frequency-dependence of ITD in several acoustical datasets. (**A**) ITD vs frequency for sound directions on the horizontal plane (azimuth 15–90°, spaced by 15°), measured in a live cat and previously reported (*Roth et al., 1980*). (**B**) Acoustical measurements on a taxidermist model of a cat in a large anechoic room (same azimuths). Dashed curves show symmetric positions for sources to the back of the animal. Note that the head is tilted to the right; azimuths are relative to the head (not the body). (**C**) Numerical calculation of ITDs by boundary element method (BEM) simulation on a 3D model of the same cat as **B** (grey shape), obtained from photographs. (**D**) Same as **C**, but with a straightened head. (**E**) Analytical calculation of ITDs for a spherical rigid head. (**F**) Same as **E**, but with an additional reflection from the ground. Head and source are placed 1.7 meter from each other and 20 cm above the ground.

The following figure supplement is available for figure 2:

**Figure supplement 1**. Envelope and fine-structure ITDs.

---

We analyzed the frequency-dependence of ITD in the acoustical recordings in the same way as we analyzed the frequency-dependence of best ITD in cells. For each azimuth and center frequency, we extracted a CD and CP from the acoustical data. For example, for a 70° azimuth and center frequency of 800 Hz (*Figure 3B*, bottom), we approximate the interaural phase difference (IPD = ITD × f) for that location by a linear function of frequency (*Figure 3C*, purple line). The acoustical CD and CP are the slope and intercept of the linear regression. Other examples are shown on *Figure 3—figure supplement 1*. Physically, the acoustical CD is the envelope ITD and the acoustical CP is the difference between envelope and fine structure ITD, expressed in cycles (see *Figure 2—figure supplement 1*). We then produced statistics by sampling center frequencies according to the CF distribution in the recorded neurons, and azimuths according to a uniform distribution in the contralateral hemifield (including front and back).

In agreement with the physiological data, the acoustic measurements show a unimodal and broad CP distribution, with a small but significant positive bias (*Figure 4A*). Consistent with the measured neurons, the acoustic CDs are mainly positive and mostly within 350 µs, but with a sizeable number of data points with negative or with large CDs (*Figure 4B*, and *Figure 4—figure supplement 3* for negative CDs). While some large neural CDs (>500 µs) lie outside the range of ITDs (*Figure 2*), all remain inside the range of acoustical CD. Finally, the acoustic data also show an inverse correlation between CD and CP (*Figure 4C*). Thus, key properties of neural CD and CP, which are the main metrics that have been used in the description of tuning to ITD, are well-matched to binaural acoustics studied with these same metrics.

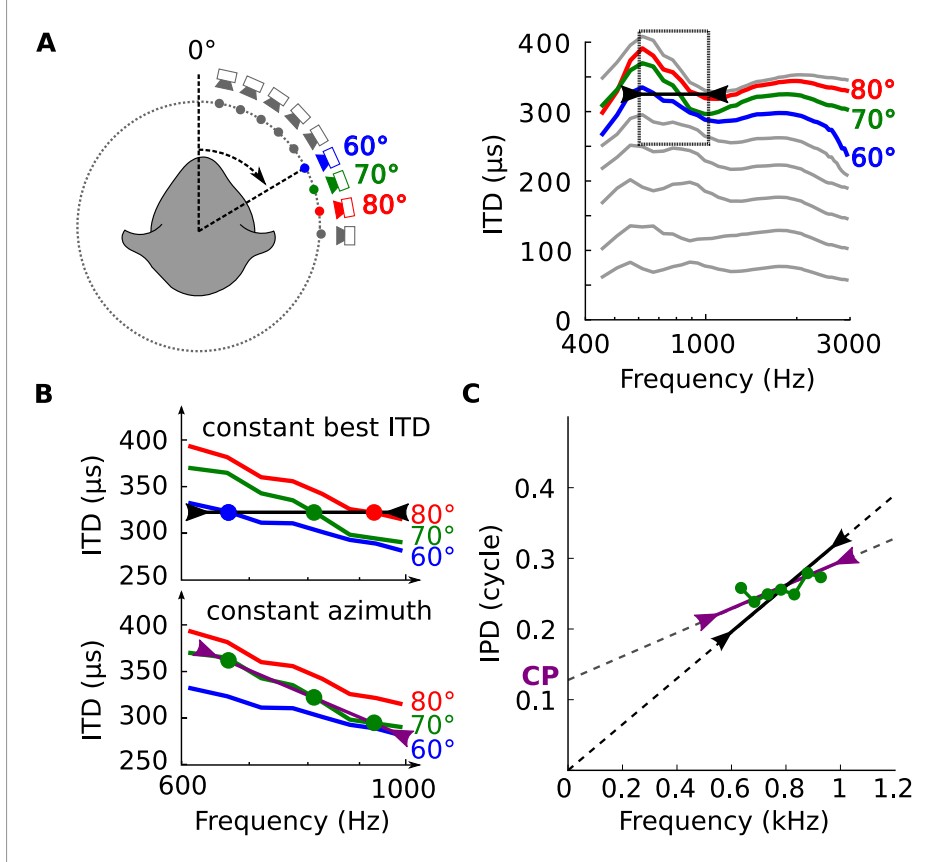

**Figure 3**. Tuning to frequency-dependent ITDs. (**A**) Left, head-related transfer functions (HRTFs) are measured binaurally for different speaker positions. Right, ITD vs frequency in cat at 60, 70 and 80° on the horizontal plane. (**B**) A neuron for which best ITD is fixed across frequency (top, black line) is tuned to different azimuths depending on frequency, while a neuron with fixed azimuth tuning has a frequency-dependent best ITD (bottom, purple line). (**C**) IPD vs frequency at 70° over a 300 Hz window around 800 Hz (green curve and circles). The black segment represents an ITD of 325 μs that is fixed across frequency, equal to the ITD at 800 Hz. The purple segment represents the best linear approximation of IPD around that frequency (intercept 0.12 cycle, slope 167 μs).

The following figure supplement is available for figure 3:

**Figure supplement 1**. IPD vs frequency for six different directions, around 650 Hz and 1600 Hz, with circular-linear fits.

Quantitatively, the distributions of CD and CP in the acoustic data depend on how space is sampled, which could alternatively be uniform over directions in the front only (*Figure 4—figure supplement 1A*), or inferred from the electrophysiological recordings (*Figure 4—figure supplement 1B*), or biased towards the side (*Figure 4—figure supplement 1C*) or the center (*Figure 4—figure supplement 1D*), as suggested in the barn owl (*Fischer and Peña, 2011*). In particular, the proportion of negative CDs varies between 8% and 31% depending on the choice of azimuth distribution, because negative CDs are observed mostly for azimuth near 0° or 180° (*Figure 4—figure supplement 3*). Distributions of CD and CP also quantitatively depend on the frequency range of the analysis (CF < 1 kHz in *Figure 4—figure supplement 2A–C*; CF > 1 kHz in *Figure 4—figure supplement 2D–F*). However, despite the quantitative differences, the same qualitative features remain. Other factors may contribute quantitative variations, such as posture, reflections off the ground, distance and elevation. Thus, the statistics of frequency-dependent ITDs in acoustical recordings qualitatively match those of frequency-dependent neural tuning to ITD.

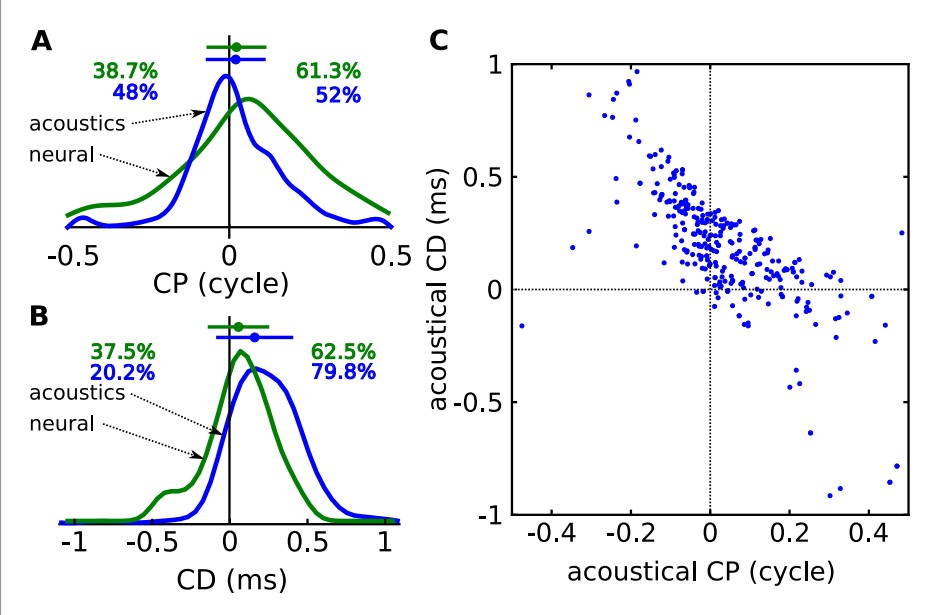

**Figure 4**. Acoustical analysis. (**A**) Distribution of CP in the cells (green) and in the acoustics based on acoustical measurements (blue). Error bars represent the mean ± STD/2, and percentages the proportion of positive/negative values. (**B**) Distribution of CD in cells (green) and in acoustics (blue). (**C**) CD vs CP in the acoustics.

The following figure supplements are available for figure 4:

**Figure supplement 1**. Acoustical predictions of CD and CP distributions for various prior spatial distributions.

**Figure supplement 2**. Acoustical predictions of CD and CP distributions in low and high frequency regions.

**Figure supplement 3**. Negative acoustical CDs.

## Theoretical explanation

The inverse correlation between CP and CD can be explained by the variation of ITD with frequency. In a simple spherical head model (*Figure 5A*), ITDs are larger at low frequency than at high frequency, but these variations are small on a local scale (*Figure 5B*, blue [*Kuhn, 1977*]). However, variations appear on a local frequency scale as soon as features of ecological environments are introduced, such as diffraction on the complex shape of head and body of real animals (*Figure 3*), or early reflections off the ground (*Figure 5B*, green [*Gourévitch and Brette, 2012*]). As a result of these variations, the IPD vs frequency curve is non-linear (*Figure 5C,D*). When a tangent is moved along this curve (dashed lines in *Figure 5C*), the slope decreases at the same time as the intercept increases. Because slope and intercept correspond to acoustical CD and CP, this means that for neurons tuned to the same spatial configuration but different frequencies, CD and CP should be inversely correlated (*Figure 5E,F*). These variations in CD and CP across frequency are small for a simple spherical head (*Figure 5F*), but they become large as soon as a ground plane is included (*Figure 5E*). We note that reflections off the ground cannot be temporally separated from the direct signal because delays are very short (about 150 μs for a source 1.5 m away from the cat's head [*Gourévitch and Brette, 2012*]) and must thus be considered as part of the signal reaching the two ears. Thus the acoustical space encountered by the animal cannot be adequately described by fixed ITDs. As a result, it cannot be unambiguously represented by neurons tuned to only fixed ITDs (corresponding to the vertical line CP = 0 in *Figure 5E,F*).

## A possible physiological mechanism

In the textbook model of ITD processing (*Jeffress, 1948*; *Joris and Yin, 2007*), a binaural neuron in the MSO detects coincidences between spikes produced by monaural neurons driven by the left and

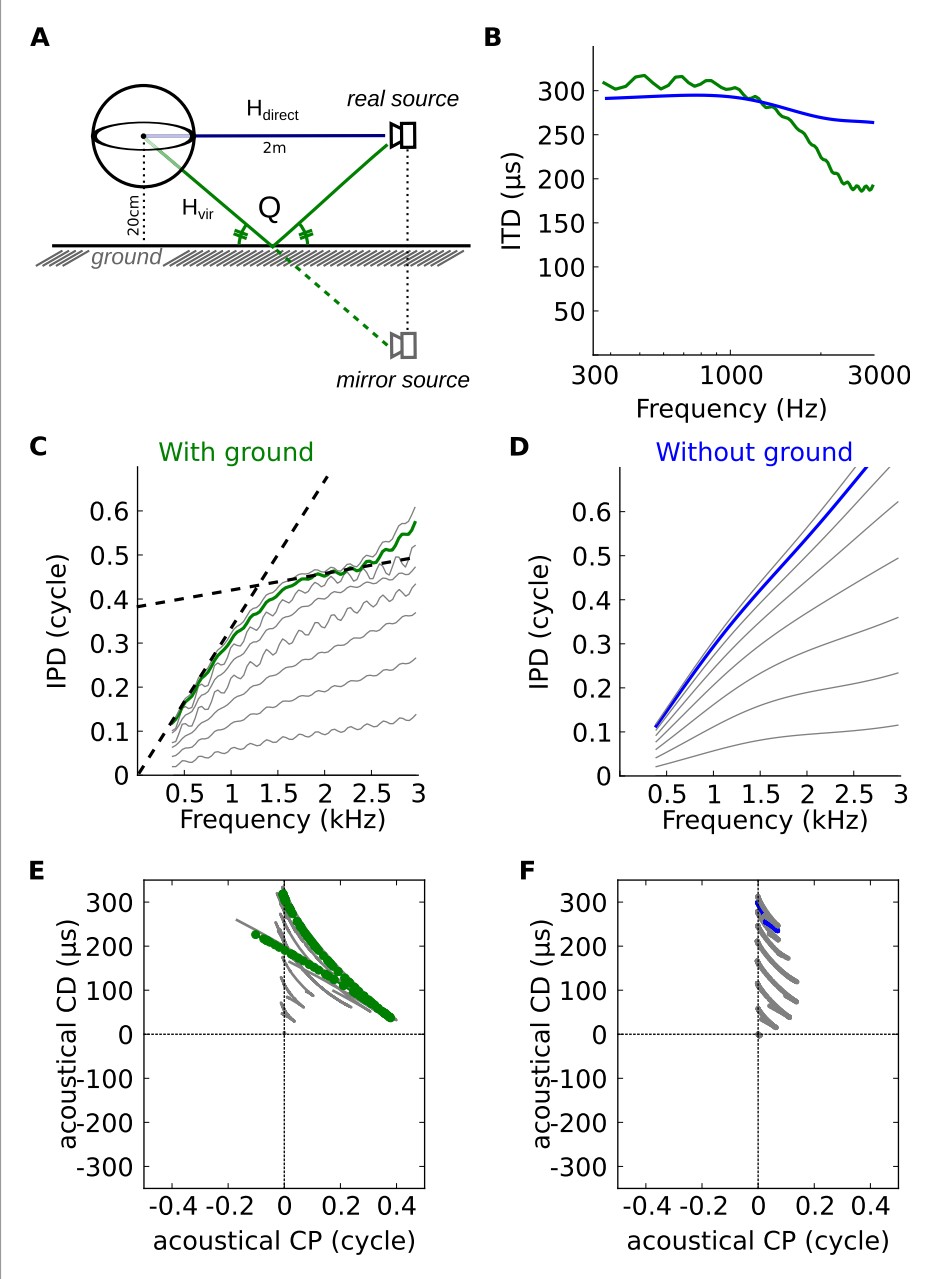

**Figure 5.** Theoretical explanation of inverse CP-CD relationship. (**A**) A spherical head model with a ground reflection. (**B**) ITD vs frequency in the spherical model for a source at 70°, with (green) and without (blue) ground reflection. (**C**, **D**) IPD vs frequency for the same position as in **B** (green and blue) and for other positions between 0 and 90° (light gray curves). (**E**, **F**) Predicted CD vs CP for the two cases.

right ear that are tuned to the same CF. Best ITDs near 0 ms imply that the left and right signals arrive coincidentally at the binaural neuron; ITDs >0 ms imply that the inputs from the contralateral ear reach the binaural neuron with some delay relative to those of the ipsilateral ear. However, this mechanism produces frequency-independent best ITDs, that is, CP = 0, which is not consistent with most of the physiological data (*Figure 1E*) (*Kuwada et al., 1997*).

Frequency-dependent best delays could be produced by small mismatches in the CFs of the monaural inputs to a binaural cell (*Schroeder, 1977*; *Shamma et al., 1989*; *Bonham and Lewis, 1999*), and some features of binaural responses are consistent with such mismatches (*Joris et al., 2006*;

*Day and Semple, 2011*). We studied the effects of mismatches in CF with a coincidence analysis of responses of several hundred cat auditory nerve fibers. We model the response of a binaural coincidence detector neuron receiving inputs from two slightly different points on the cochlea, leading to a CF mismatch (*Figure 6A*, top panel). This is achieved by counting the coincidences between the spike trains of two recorded fibers with slightly different frequency tuning (*Figure 6A*, right panel), in response to a range of pure tones. By varying the delay between the spike trains, 'pseudobinaural' ITD curves at different frequencies are generated.

*Figure 6C* shows the results of coincidence analysis on responses to tones with frequencies between 400 and 2200 Hz (bin width = 50 µs), for two fibers with similar but slightly different frequency tuning (CF = 1092 Hz and 1133 Hz, *Figure 6B*). The maxima of these ITD-curves are the best ITD of a model binaural coincidence detector receiving inputs from those two fibers: they show a frequency dependent BD with CP = 0.234 and CD = −0.194 ms (*Figure 6D*).

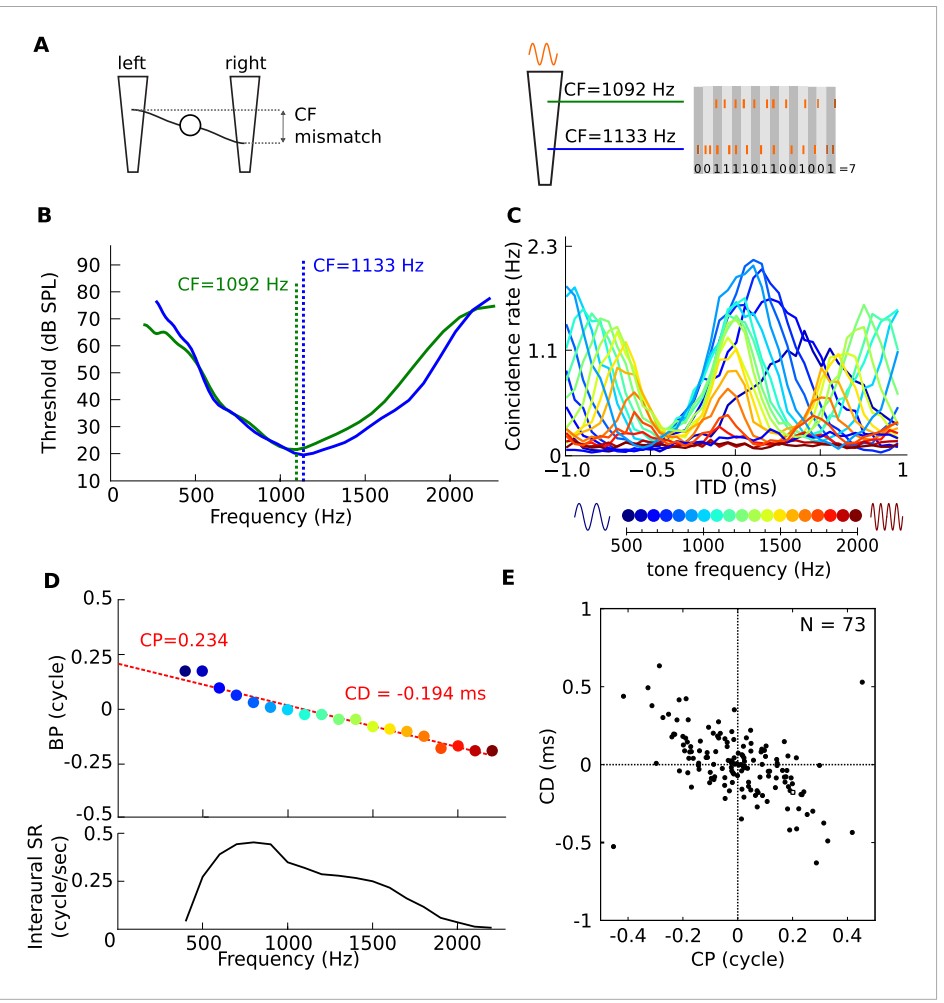

**Figure 6**. Mechanism for frequency-dependent neural tuning. (**A**) Schematics of the coincidence analysis. The left schematic illustrates the concept of cochlear disparities. The trapezoids schematize the cochlear basilar membrane. A left and right fiber originate from a different cochlear place and converge on a binaural neuron. The right schematic illustrates the counting of coincidences between spike trains from two fibers in response to a single tone. Due to the cochlear traveling wave, the spike trains of the more apical (green) fiber are expected to be delayed in time and lagged in phase relative to the more basal (blue) fiber. (**B**) Threshold tuning curves of the two example fibers. (**C**) Pseudobinaural tuning curves: Coincidence counts as a function of ITD for a pair of fibers for different tone frequencies. Each curve is color coded with the frequency of the stimulus, scale is presented below the plot. (**D**) BP as a function of frequency for the same nerve pair as in **C**. (**E**) CD vs CP over a population of coincidence detectors receiving inputs from cat auditory nerve fibers with mismatched CF (<0.1 octave; CF < 3.3 kHz).

Using auditory nerve data from a single animal, we simulated the CP and CD of a population of binaural coincidence detector cells. We counted coincidences for different delays between spike trains of 73 pairs of fibers with slightly mismatched CF (≤0.1 octave), and processed the resulting coincidence counts with a CP-CD analysis identical to that used on real binaural neurons (*Figure 1*). *Figure 6E* shows that here as well, CP is broadly distributed and inversely correlated with CD. Note that the CP distribution is centered on 0 because we symmetrized the distribution by representing each fiber pair twice to simulate random mismatches between the inputs from the two sides, where sometimes the ipsilateral fiber is higher in CF and sometimes the contralateral fiber (i.e., reflecting both positive and negative CF mismatches). The distribution of *Figure 6E* is consistent with the phase characteristics of the cochlear traveling wave, which generates frequency-dependent delays and ultimately drives the hair cells and auditory nerve fibers (*Schroeder, 1977*; *Shamma et al., 1989*; *Bonham and Lewis, 1999*; *Day and Semple, 2011*), and shows that very small CF mismatches are sufficient to produce CPs of the same magnitude as measured in binaural cells.

In order to provide direct evidence of CF disparities in mammals, we obtained in vivo patch clamp recordings from 6 gerbil MSO cells, using the method described in a recent study (*Franken et al., 2015*). The rate of excitatory presynaptic events (EPSPs) was measured during monaural ipsi- or contralateral presentation of pure tones at different frequencies (*Figure 7*). The data show that afferents to MSO cells can differ in their spectral composition, albeit in a complex manner. This confirms an observation in juxtacellular recordings (see *Figure 5* of (*van der Heijden et al., 2013*)) and suggests that CF mismatches in mammals may play a role in shaping tuning of ITD sensitive cells.

Random cochlear disparities are not a sufficient mechanism to account for the physiological distribution (*Figure 1E,F*) because they fail to account for the positive bias of CP and CD values. Such a bias can be obtained by systematic cochlear disparities (i.e., where the contralateral inputs are tuned to lower CFs than the ipsilateral inputs) (*Joris et al., 2006*). It can also be obtained by considering mismatches in axonal conduction delays, in addition to CF mismatches: axonal delays add to the CD without changing the CP. Such delay mismatches could result from the longer contralateral than ipsilateral path length to reach the off-midline MSO, or from other structural axonal differences between contra- and ipsilateral branches (*Seidl et al., 2010*).

## Discussion

We analyzed the frequency-dependence of ITD in acoustical recordings, and found that its statistical properties qualitatively match those of ITD tuning in binaural neurons of the cat's IC. This is consistent with the hypothesis that these neurons are tuned to binaural features of ecological acoustic space rather than to a fixed ITD. Moreover, we find that the binaural tuning observed does not need complex wiring but could be based in a rather straightforward way on properties of the auditory periphery.

### Ecological interaural time disparities, and constraints on binaural sensitivity

The binaural tuning in our sample of IC neurons, quantified with traditional measures of CD and CP, matches that of many previous physiological studies (*Palmer and Kuwada, 2005*). However, we go further by making an explicit comparison with distributions of the same metrics applied to acoustical measurements. This leads to two new insights. First, we provide evidence that the distribution of neural CDs and CPs and acoustical CDs and CPs are similarly constrained. Second, large CDs are present acoustically. Taken together our results suggest that the binaural tuning of IC cells is constrained by the range of delays that the animal experiences.

Our findings and interpretation relate to two points that have been much discussed in the literature. Non-zero CPs in the neural data have puzzled investigators since the first reports of ITD-sensitivity, both in terms of their physiological origin and their functional significance (*Rose et al., 1966*; *Yin and Kuwada, 1983*). Perhaps even more puzzling has been the discrepancy observed between the distribution of physiological CDs and acoustical (phase) ITDs, for example, as reported in HRTF measurements (for cat: [*Roth et al., 1980*; *Tollin and Koka, 2009*], and *Figure 2*). As was pointed out since the first physiological data became available (*McFadden, 1973*), neuronal CDs seem to cover an 'unnecessarily' wide range including ITDs that animals will not naturally encounter. Various interpretations have been given to this discrepancy (*McAlpine et al., 2001*). However, the appropriate comparison is not between physiological CDs and acoustical phase ITDs, but with acoustical CDs (i.e., envelope ITDs). As shown here for acoustical measurements (*Figure 4B*), due to

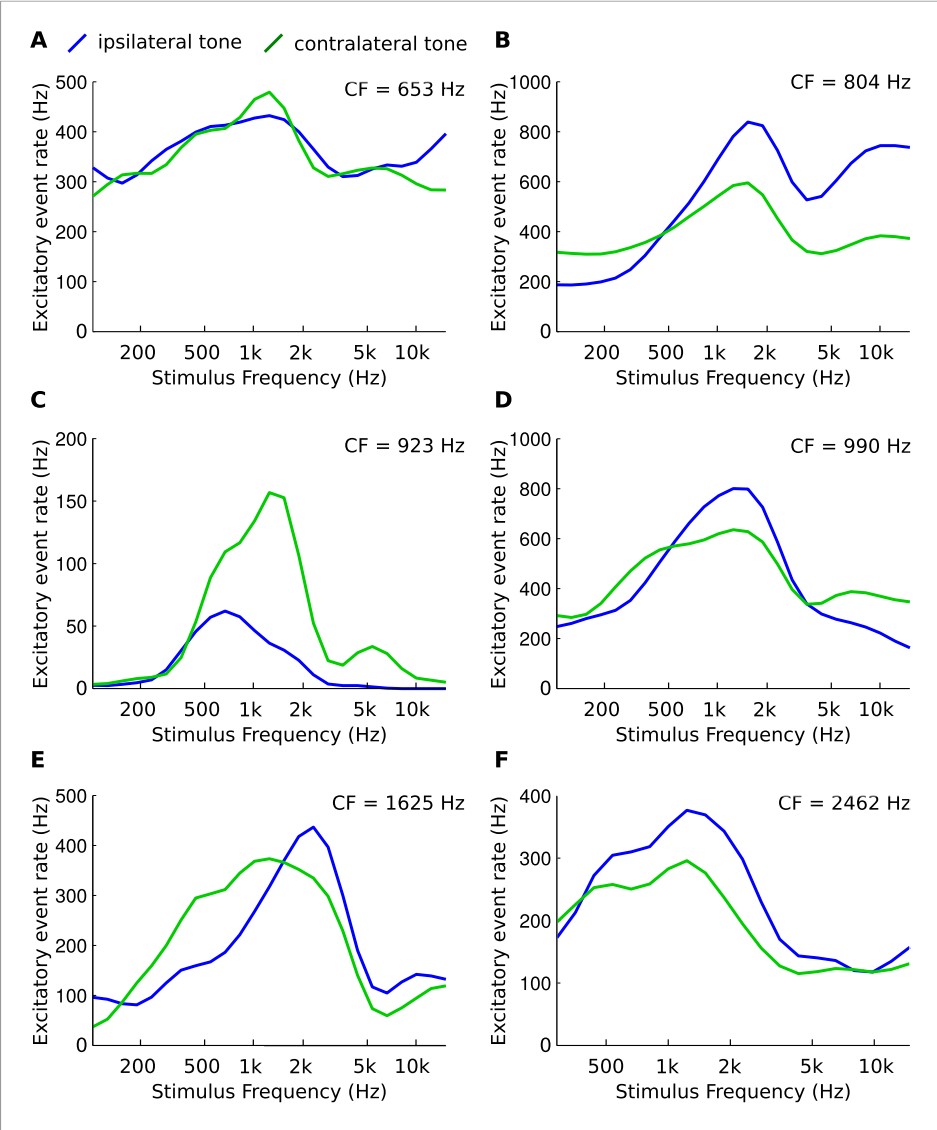

**Figure 7**. Asymmetries in frequency tuning in the excitatory inputs to the gerbil medial superior olive (MSO). (**A–F**) Rate of excitatory presynaptic events (EPSPs) in 6 MSO cells, with different CF, in response to tones as a function of frequency. Stimuli are presented ipsilaterally (blue function) or contralaterally (green function). Only EPSPs ≥ the median EPSP amplitude are included. Functions were smoothed using a 3-point running average.

the frequency dependence of the phase ITD, CDs have a wide distribution and actually exceed the range of physiological CDs. The range of acoustical CDs that drives a given cell can therefore exceed the range of phase ITDs computed from HRTF data.

## Complexity of ecological acoustical environments

Because the frequency-dependence of ITD reported here reflects the physical interaction of the sound with the ears, head, body, and ground plane, it should also apply to other species. We applied our analysis to HRTFs of humans (*Figure 8A–C*), for whom ITDs dominate below ~1.5 kHz (*Wightman and Kistler, 1992*; *Macpherson and Middlebrooks, 2002*). The same basic relationship between acoustical CD and CP is present.

Our study emphasizes the notion that the binaural cues that an auditory system has to process in ecological environments are much more complex and rich than in idealized settings. We have focused here on the contribution of sound diffraction by the head and body, a phenomenon that is always

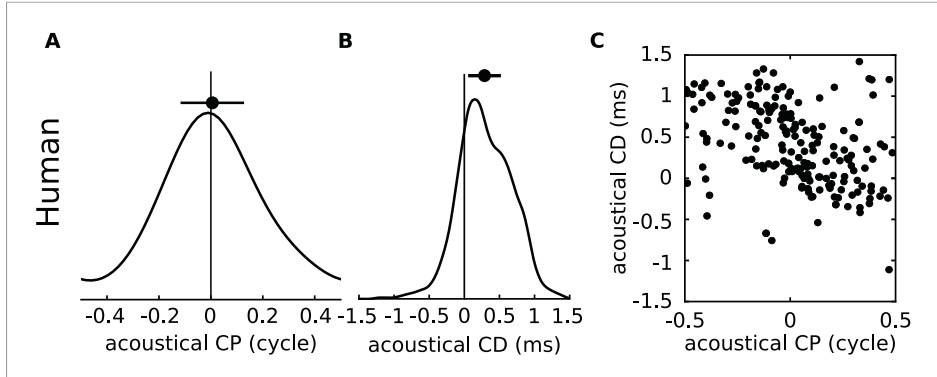

**Figure 8**. Acoustical analysis of human. (**A**–**C**) Predictions using human HRTFs (*IRCAM LISTEN HRTF database*) for the distributions of acoustical CP (**A**), CD (**B**) and 200 sample points from the joint CP-CD distribution (**C**) over a 100–1500 Hz frequency range.

present, even in anechoic environments. Because the effect depends on the detailed morphology of the animal, it varies with posture in a systematic way (see *Figure 2C,D* and [*Rébillat et al., 2014*]). Early reflections, in particular from the ground (again a constant element in ecological environments of terrestrial animals), produce interferences with the direct sound that result in large frequency-dependent variations of ITD, especially at low frequencies (*Gourévitch and Brette, 2012*). Ground reflections typically arrive very shortly after the direct sound and are therefore an integral part of the binaural signal received by the animal. The interferences they produce are determined primarily by the delay of the reflection (related to the distance) and secondarily by the nature of the reflecting object (grass, sand, snow, etc). There are many other sources of complexity in ecological environments. For example, natural sound sources are rarely point sources producing spherical wavefronts, as assumed in most studies. Many are large (a river) and directional (human speech), some are partially occluded by objects (a prey hiding in a bush). The sound of one's own footstep travels through air but also through the body, which produces different acoustical cues (*Hood, 1962*). We suggest that the binaural system of animals is adapted to these complex natural acoustical cues.

This complexity must be addressed by any sound localization theory and imposes particular difficulties (*Brette, 2010*; *Goodman and Brette, 2010*; *Goodman et al., 2013*). For example, if ITD depends on the position of the pinnae, or posture, as is the case for the cat (*Figure 2*, and [*Tollin and Koka, 2009*]), then proprioceptive information must be taken into account to interpret binaural cues. Unarguably, this property makes the task more difficult for the animal, but not more difficult than taking into account other known causes of ITD variation—such as source distance and elevation—when mapping ITD to source direction.

The fact that ITD depends on factors other than azimuthal position implies that binaural neurons in the MSO are not tuned to source direction per se, but rather to temporal binaural features of acoustical spaces—a notion that generalizes frequency-independent ITD. If binaural tuning develops from exposure to natural sounds (*Seidl and Grothe, 2005*), then it would be expected that its properties reflect those of ecological acoustical environments, especially given that very small cochlear disparities give rise to significant frequency-dependence of ITD tuning in binaural neurons (*Figure 6*).

We can think of two ways of testing the hypothesis that binaural neurons are tuned to properties of ecological acoustical spaces. One is to raise animals in environments with structured but manipulated acoustical cues (as opposed to unstructured noise as in [*Seidl and Grothe, 2005*]), for example using earplugs, and observe the changes in ITD tuning of binaural neurons. Another one is to measure spatial receptive fields of binaural neurons (instead of ITD selectivity curves) and test whether they depend on spectral properties of sounds.

## Mechanisms of ITD tuning

In previous work, nonlinear phase-frequency relationships (i.e., where the best ITD is not constant) are typically considered to be 'biological noise'. Thus, even though frequency-dependent best ITDs are clearly an important characteristic of the physiological data (*Figure 1*), there are to our knowledge no

functional interpretations of this characteristic. The observation (*Figure 6*) that CF mismatches can generate nonlinear phase-frequency relationships (CP ≠ 0), combined with the presence of such relationships in the binaural and acoustic data, suggest a rather simple new view on neural binaural properties. The view is that limitations in the accuracy of wiring in the binaural system produce a property that benefits binaural hearing. Refinements of this wiring in combination with mismatches in axonal delays produce a range of binaural sensitivities well-matched to the acoustic scenes that the system is faced with. It was previously shown that random CF mismatches in the wiring to the contralateral and ipsilateral pathways to IC could account for the negative correlation between CF and range of best ITDs measured with delayed noise (*Joris et al., 2006*). Here we show that appropriate CF and axonal delay mismatches can account for the dependence of best ITD on tone frequency within single cells, matching acoustical properties.

Besides cochlear disparities, a range of mechanisms have been proposed to account for the best delays of ITD-sensitive neurons: axonal length (*Jeffress, 1948*), phase-locked inhibition (*Brand et al., 2002*), asymmetric placement of the axon (*Zhou et al., 2005*), asymmetry in synaptic kinetics between contra- and ipsilateral inputs (*Jercog et al., 2010*), differences in axonal conduction time (*Seidl et al., 2010*), and phase delays generated by the interaction of intrinsic properties with input spike patterns (*Franken et al., 2015*). All of these proposals face difficulties to explain all the data available, and it is at present unclear which of these mechanisms, or perhaps mix of mechanisms, is in place, or whether perhaps the main mechanism has not been identified yet. Moreover, discussions of these various mechanisms usually focus on CD, leaving it unclear whether and how they would affect CP. Axonal length and conduction time are expected to generate pure time delays and would therefore not generate CPs different from 0. The report first proposing phase-locked inhibition as a source of internal delay (*Brand et al., 2002*) provided model results illustrating how best ITD at different frequencies was little affected by stimulus frequency, that is, also predicted that inhibition would be equivalent to a pure time delay (CP = 0). Later, more extensive modeling (*Day and Semple, 2011*) showed that adjustment of model inhibitory parameters allows creation of a wide range of non-zero CP values, but that CP remained within 0.1 cycle when more realistic inhibitory synaptic time constants were used. To our knowledge, asymmetrical axonal placement and asymmetry in synaptic kinetics have not been examined regarding a possible contribution to CP. However, these two proposals, as well as phase-locked inhibition, received little experimental support from a recent in vivo intracellular MSO study (*Franken et al., 2015*).

Cochlear disparities have been proposed before as a mechanism for generating internal delays, in place of axonal delay mismatches (*Schroeder, 1977*; *Shamma et al., 1989*; *Bonham and Lewis, 1999*). The original form of that hypothesis, where such disparities are the sole source of internal delays, has been rejected in the barn owl: CF mismatches are observed but they are relatively small and do not correlate with ITD tuning (*Pena et al., 2001*; *Fischer and Peña, 2009*). They are nonetheless significant as they contribute predicted delays of up to 50 µs (*Pena et al., 2001*). Importantly, in contrast with the original cochlear disparity hypothesis, our proposed mechanism combines small CF mismatches (just 40 Hz in *Figure 6*) and axonal delay mismatches. A mix of cochlear disparities and pure time delays was also proposed to account for non-linear phase-frequency relationships observed in gerbil MSO responses (*Day and Semple, 2011*). In cat MSO, (*Yin and Chan, 1990*) reported that monaural best frequencies differed by 0.2 octaves or less for 13 of the 18 cells (and more for 5 cells). As we show in *Figure 6*, this order of magnitude is sufficient to produce the observed frequency-dependence of best ITD. Intracellular recordings from MSO neurons allowed us to directly measure the monaural inputs and confirm that they can differ in spectral properties. Additional experiments are needed to compare frequency-dependent properties of ITD tuning with the mismatched frequency tuning of monaural inputs.

Our data are from the IC, one synapse removed from the sites of binaural interaction. In our mechanistic explanation, we have assumed that frequency-dependent properties of ITD tuning observed in IC neurons are inherited from the MSO. This is in agreement with electrophysiological data in the MSO of cats (*Yin and Chan, 1990*) and gerbils (*Day and Semple, 2011*; *van der Heijden et al., 2013*), which also show frequency-dependent ITD tuning (broad distributions of CP). Alternatively, nonlinearities at the level of the IC could reflect convergence of inputs, for example, from binaural neurons in MSO, LSO, and DNLL (*McAlpine et al., 1998*). For example, LSO neurons have CP close to 0.5 (*Joris, 1996*; *Tollin and Yin, 2005*). Neurons with CP close to 0 have been categorized as 'peakers', because peaks of their ITD selectivity curves align across frequency, while

neurons with CP close to 0.5 have been categorized as 'troughers', because troughs align across frequency (*Batra et al., 1997*). These two categories are traditionally considered as predominantly reflecting MSO or LSO input, respectively. Neurons with intermediate CP are then categorized as 'tweeners', presumably reflecting a combination of MSO and LSO inputs. However, our electrophysiological data showed no clear categories in the distribution of CP (*Figure 1E*). Instead, the distribution was broad and unimodal around CP = 0, which argues against the categorization of cells as peaker, trougher, and tweener. In addition, electrophysiological studies in the MSO of cats (*Yin and Chan, 1990*) and gerbils (*Day and Semple, 2011*) also show cells with non-zero CP. Our auditory nerve data shows that very small mismatches in CF of inputs to binaural cells are sufficient to produce significantly non-zero CP, without postulating any additional mechanism than coincidence detection (*Figure 6*).

In summary, while the mechanisms of ITD tuning remain unclear, one attractive feature of cochlear disparities is that they provide a simple mechanism to generate non-zero CPs, which we show here are a likely desirable property of the binaural system as they match the acoustics that animals face, when combined with a physiological source of pure time delays.

## Materials and methods

### Electrophysiological recording

Our methods for single unit recording have been described before: in the cat IC and auditory nerve (*Joris et al., 2005*, *2006*), and in the gerbil MSO (*Franken et al., 2015*). All procedures were approved by the institutional Animal Care Committee and were in accordance with the NIH Guide for the Care and Use of Laboratory Animals.

In cat experiments, anesthesia was induced with acepromazine and ketamine and maintained for surgical preparation and recording with pentobarbital. Induction of anesthesia in gerbils was with ketamine and xylazine; maintenance was with ketamine and diazepam. All animals were placed on a heating pad in a double-walled sound-attenuated chamber. Sound stimuli were delivered dichotically with speakers coupled to earbars that were tightly coupled to the ear canals. The stimuli were generated digitally and were compensated for the acoustic transfer function measured with a probe microphone near the eardrum.

In the cat, bullas were vented with tubing. The IC was exposed anterior to the tentorium; the auditory nerve was exposed via a posterior fossa approach. Single IC neurons were recorded with metal electrodes; auditory nerve fibers with high impedance glass micropipettes. The neural signal was amplified, filtered, timed (1 μs resolution) and displayed using standard techniques. The dorsal border of the central nucleus of the IC was defined physiologically by the presence of background discharges phase-locked to binaural beats of low-frequency pure tones, and the IC was histologically processed to confirm the site of recording to the central nucleus. Binaural IC recordings were obtained from 31 animals, monaural auditory nerve recordings from 1 animal.

Binaural beat stimuli were long duration (typically 1 or 5 s) tones presented over a range of frequencies bracketing the limits of the response area; the step increment was between 25 and 200 Hz to ensure adequate sampling. The tones to the two ears had a small (1 or 2 Hz) difference. Typically the contralateral ear was at the higher frequency (positive beat) but the opposite (negative beat) was also often tested. The number of repetitions was typically between 1 and 10, and the SPL was 60 dB.

In vivo whole-cell recordings were obtained from MSO neurons in the gerbil (*Franken et al., 2015*). Membrane potential was recorded in current clamp mode during monaural presentations of pure tones at different frequencies (typically 1–3 repetitions of 50–250 ms long tones in 0.3 octave increments, at 60 dB SPL). Excitatory post-synaptic events were detected as described in *Franken et al. (2015)*.

In both experiments, CF to binaural stimulation was determined with a threshold tracking algorithm.

### Analysis of cat IC data

#### Selection

The spikes fired by a cell in response to multiple presentation cycles of a binaural beat are aggregated into a period histogram. The average vector of the period histogram has length defined as the vector strength (VS, [*Goldberg and Brown, 1969*]) and angle defined as the BP. There were

on average 9.5 (±4) different tone frequencies per cell, and never fewer than 5. At each frequency, interaural phase difference (IPD) sensitivity was tested using a Rayleigh test. Only data points where the null hypothesis of uniformity could be rejected were included (p < 0.001). If a strong onset component was present, this part of the response, or the response to the entire first beat cycle, was discarded.

## Calculation of CP and CD

For each cell, the relationship between frequency and BP is fitted to a line in phase space, which yields a phase intercept called the CP and a slope called the characteristic delay (CD):

$$BP(f) \approx CP + CD.f,$$

where f is frequency and phases are in cycle. Because phase space is circular (0 cycle and 1 cycle are the same phase), the proper way to fit a line is to perform a circular linear fit, involving circular distances (*Luling et al., 2011*). This avoids unwrapping the BP, which is unreliable.

Additionally, because the cell does not respond in the same way at all frequencies, some data points should be given a stronger weight in the regression. This weight is generally chosen to be the *sync-rate* (SR), which is the product of discharge rate and vector strength (VS, defined with respect to the beat frequency [*Yin et al., 1986*; *Kuwada et al., 1987*]).

The regression then consists in determining the values CP and CD that minimizes the following quantity:

$$\sum_f SR(f).d(BP(f), CP+CD.f),$$

where d(.) is the circular distance between 0 and 1:

$$d(x,y) = \frac{1}{2}[1 - \cos(2\pi(x-y))].$$

Because this minimization problem is non-linear and potentially has local minima, it is solved in two steps. First, the (CP, CD) parameter space is systematically sampled on a regular grid (CP between −0.5 and 0.5 cycle, CD between −2 and 2 ms). The best pair of parameter values is then used as the initial value to a gradient descent algorithm.

## Range of best ITD

Best ITDs are computed over a frequency range where the response's *sync-rate* (SR, see above) is above 80% of its maximal value (*Figure 1B*). The range of best ITDs (*Figure 1C*) is the difference between the maximal and minimal best ITD over this 80% range.

## Fit quality

The quality of the circular linear regression described above was determined in two ways (*Figure 1—figure supplement 1*). First, we calculated the residual error, which is the distance between data points and the regression line:

$$residual = \sqrt{\frac{1}{N}\sum_f d(BP(f), CP+CD.f)},$$

which gives a number between 0 (perfect match) and 1. The distribution of residual errors from the cell data and the model data are reported in *Figure 1—figure supplement 1A*.

Second, we checked the statistical significance of the fit using the same approach as in *Yin and Kuwada (1983)*. We want to reject the null hypothesis that the cells have uniformly distributed BP between −0.5 and 0.5 cycles, using the residual error as a statistic. We generate surrogate data under the null hypothesis, the only parameter of interest being the number of measured frequency points. We then perform the circular linear fit and measure the residual error. We obtain the probability distribution of observing a given residual error under the null hypothesis, for every possible number of frequency points. The cumulative of that distribution gives the linearity significance measure (*Figure 1—figure supplement 1B*). The null hypothesis could not be rejected (p > 0.05) in only 1.2% of the cells. Example of significant (green) and non-significant (red) circular-linear fits are presented for cell BP (*Figure 1—figure supplement 1C*) and acoustical IPD data (*Figure 1—figure supplement 1D*).

## CP-CD correlation analysis

CP and CD are two covarying quantities obtained from a single linear regression, therefore spurious correlations across the CP and CD measured in different cells could be induced. We designed a statistical test to check that this could not by itself explain the inverse correlation between CP and CD observed across all cells (*Figure 1—figure supplement 2A*). In a linear regression, the errors in estimated intercept and slope are inversely correlated (*Figure 1—figure supplement 2B*). Despite this, the correlation across CP and CD measurements for different cells can be either negative or positive (*Figure 1—figure supplement 2C* shows an example of positive correlation for 4 cells).

We try to reject the null hypothesis that CP and CD are independent *across the population of cells*, despite the covariation of CP and CD estimates *within each cell*. We first estimate the covariation of CP and CD estimates by computing the (CP, CD) distribution for each cell obtained from bootstrapped samples (*Figure 1—figure supplement 2B*). That is, if there are n measured frequencies, we pick n frequencies at random (with possible repetitions) and measure CP and CD, then reiterate many times to obtain a distribution, from which we extract the covariance matrix.

Second, we generate 200 random (CP, CD) points under the hypothesis that CP and CD are independent, using the distributions measured in cells (*Figure 1—figure supplement 2D*). Correlated noise is then added to each datapoint, according to the distribution previously measured in bootstrap samples (*Figure 1—figure supplement 2E*). Correlation is then measured across all data points using Spearman's rank correlation ρ (which does not require linearity, *Figure 1—figure supplement 2F*). The procedure is repeated $10^6$ times so as to obtain a distribution of ρ, which is slightly biased towards negative values (*Figure 1—figure supplement 2G*), reflecting the negative correlation in the added noise. Comparing the measured correlation in the original data to the cumulative of the distribution on the surrogate data provides us with a p-value of $10^{-6}$, thus we can safely reject the hypothesis that the observed CP-CD correlation is due to the co-variation of linear regression estimates.

## Acoustical measurements

HRTFs of an anesthetized cat were obtained from a previous study (*Tollin and Koka, 2009*). They consist of 36 measurements in the horizontal plane with evenly spaced azimuth.

The analysis was also performed on other HRTF sets (*Figure 2*). We measured HRTFs of a taxidermist model of cat from the Paris Museum of Natural History (*Figure 2B*) in a large anechoic chamber at IRCAM (Paris). We used the same experimental setup as the (*IRCAM LISTEN HRTF database*, date unknown) using the sine-sweep method. Miniature microphones were placed at the entry of the meatus, which had been occluded by the taxidermy procedure.

As a control, we obtained a 3D model of the same cat from photographs (*Figure 2C*, insert) and numerically calculated HRTF with a boundary element method (*Otani and Ise, 2006*; *Rébillat et al., 2014*). The calculations were also performed on the 3D model after manually tilting the head 45° on the 3D model so as to align it with the body.

HRTFs of a spherical head model were computed based on the analytic solution of the wave equation (*Figure 2E,F*, *Figure 5*), as detailed in (*Duda and Martens, 1998*). The head diameter was measured on the 3D, model of the cat (d = 7.3 cm).

Naturalistic ground reflections were included in the spherical model (*Figure 2F*, *Figure 5*) using the method described in (*Gourévitch and Brette, 2012*). The head was placed 20 cm above ground, and the sound source was placed at the same distance of the ground, one meter away from the head. The ground was modeled with flow resistivity of $5.10^5$ (kP.s/m$^2$), which is between grass ($10^5$) and sand ($10^6$).

In *Figure 8*, we used previously measured HRTFs of one randomly picked subject of the (*IRCAM LISTEN HRTF database*), measured for 72 positions on the horizontal plane.

## Analysis of acoustical measurements

### HRTF analysis

We analyze the frequency-dependence of ITD in HRTFs in frequency bands. For each center frequency F, we consider a window centered around F with bandwidth $BW(C) = F/Q(F)$, where Q(.) was a linear function of F derived from the cell recordings by linear regression ($Q(CF) = 1.04 + (3.8 \times 10^{-4}$ s).CF). We then compute acoustical CP and CD from the IPD in this window as in recorded cells.

In *Figure 4*, azimuths are sampled from a uniform distribution in the contralateral hemifield (see below for other distributions shown in *Figure 4—figure supplement 1*). Center frequencies are

distributed as the CFs of recorded cells. We first estimate the distribution of CFs in the data using Gaussian kernel estimation, constrained between 300 Hz and 3 kHz. Then N = 200 points are drawn from this distribution, with random azimuth (as in *Figure 4C*).

## Distributions of azimuth

We also considered four alternative distributions of azimuth (*Figure 4—figure supplement 1*). In *Figure 4—figure supplement 1A*, azimuth is uniformly distributed between 0° and 90° (i.e., only in the front). In *Figure 4—figure supplement 1B*, azimuth is distributed according to the distribution of best azimuth (BA) of the cells (see below). In *Figure 4—figure supplement 1C*, azimuths near 90° are favored, according to the following (non-normalized) distribution:

$$P(\theta) = (1 + \cos(\theta - 90))^4.$$

In *Figure 4—figure supplement 1D*, azimuths near the midline are favored, according to the distribution proposed in the barn owl in (*Fischer and Peña, 2011*), with a contralateral constraint.

## BA

The BA of a cell is defined as the azimuth that elicits the largest response in that cell. We assume that it occurs when the IPD is closest to the cell's BP across the relevant frequency range. Therefore the cell's BA is the azimuth that minimizes the following quantity:

$$\sum_f d(BP(f), IPD(f)),$$

where d(.) is the circular distance. The minimization is performed as previously described. Frequency points where chosen as in the acoustical analysis.

Since BPs and IPDs are not necessarily measured at the same frequency points, the closest available frequency was chosen for IPD(f), which was never further than a few Hz away given the high sampling rate of HRTF measurements.

## Signal-processing interpretation of ITD

### ITD definitions

Consider the HRTF filters at given position with an IPD measured in cycles. There are two common definitions of interaural delays:

*group ITD* measures the ITD of the envelopes:

$$ITD_g(f) = \frac{d}{df} IPD(f).$$

*phase ITD* measures the ITD of the signals' fine structure:

$$ITD_p(f) = \langle \frac{IPD(f)}{f} \rangle,$$

where the bracket operator ⟨.⟩ represents the unwrapping operation. With a discrete set of frequency points, it means that phase jumps $|IPD_k - IPD_{k-1}|$ greater than 1 cycle are replaced by their 1-cycle complement.

If the phase ITD is constant, then the interaural phase (IPD) depends linearly on frequency: $IPD(f) = ITD_g f$ (*Figure 2—figure supplement 1A*, top). In this special case, which occurs in the absence of sound diffraction (no head), group and phase ITDs are equal at all frequencies.

In measurements and in models of HRTFs, however, the phase ITD varies with frequency (see, e.g., *Figure 2*). Equivalently, the IPD is not a linear function of frequency anymore (*Figure 2—figure supplement 1A*, bottom, gray line), and group and phase ITD are different. Over a small enough frequency band (such as within a single auditory filter), the IPD can still be correctly described with an affine approximation (*Figure 2—figure supplement 1A*, bottom, black line):

$$ITD_p(f) = IDI(f) + f\ ITD_g(f).$$

Notice on the above that if $ITD_p(f)$ is constant then IDI = 0 cycles for all frequencies. We thus define the Interaural Diffraction Index (IDI), a frequency-dependent quantity that represents how much the ITD varies because of diffraction effects:

$$IDI(f) = f\big(ITD_p(f) - ITD_g(f)\big).$$

IDI is a phase quantity (measured in cycles) that also has an interesting interpretation in terms of signal processing (see section below).

## Envelope and fine-structure ITDs

If the phase ITD is constant across frequencies, then monaural signals are delayed versions of one another. On the other hand, when phase ITD is frequency-dependent; it is unclear what transformation monaural signals go through. On *Figure 2—figure supplement 1* we simulated the effects of complex, frequency-dependent ITD on a source signal. The top panel of *Figure 2—figure supplement 1B* depicts the ipsilateral signal, an amplitude-modulated pure tone (it is equal to the source signal for simplicity). We simulate HRTFs with either a linear phase response with $ITD_g = 5$ ms, $IDI = 0$ cycles (*Figure 2—figure supplement 1A*, top) or an 'affine' phase response with $ITD_g = 5$ ms, $IDI = 0.5$ (*Figure 2—figure supplement 1A*, bottom). The signal arriving at the contralateral ear is depicted on middle and bottom panels of *Figure 2—figure supplement 1B*. Because the envelope delay is the same in both situations, the envelopes are delayed by the same amount regardless of IDI. The fine structure, however, is delayed differently in that it is in antiphase when $IDI = 0.5$. In fact, the fine structure undergoes an additional phase shift equal to the value of IDI.

This effect is reflected on the interaural cross correlation functions, where the position of the peak of the envelope of the cross correlation is the same in both situations (*Figure 2—figure supplement 1C*, green tick), while the peak of the fine structure of the cross correlation (blue tick) is shifted by an amount in phase equal to the IDI (red segment).

In conclusion, a general frequency-dependent ITD can be understood as having two effects on the monaural signals. The contralateral signal is the ipsilateral signal delayed by an amount equal to the group ITD, while the contralateral signal's fine structure undergoes an additional phase shift equal to the IDI.

## Application to the cat HRTFs

We computed $ITD_p$, $ITD_g$ and IDI over a range of frequencies on the cat HRTF (*Figure 2—figure supplement 1D*, same data as is in *Figure 2A*), for an azimuth of 70°. Distributions of $ITD_g$, $ITD_p$ and IDI over frequencies (300–3000 Hz) and positions (on the horizontal plane) are depicted in *Figure 2—figure supplement 1E*. Notice how, in general, $ITD_g$ and $ITD_p$ are different and the distribution of $ITD_g$ is wider than that of $ITD_p$.

The computation of $ITD_p$, $ITD_g$ and IDI followed the same method as the computation of CP and CD. For a given azimuth and center frequency $f$, the IPD is approximated by an affine function of frequency on a window of size $f/4$ around $f$, using circular linear regression, as previously described. The slope of the regression is an estimate of $ITD_g(f)$ and the intercept of $IDI(f)$. Phase ITD is obtained with the formula:

$$ITD_p(f) = ITD_g(f) + \frac{IDI(f)}{f}.$$

## Acknowledgements

We thank Daniel J Tollin for sharing measured cat HRTFs. We thank the Museum of Natural History for lending the taxidermist model.

# Additional information

### Funding

| Funder | Grant reference | Author |
| --- | --- | --- |
| European Research Council (ERC) | ERC StG 240132 | Romain Brette |
| Agence Nationale de la Recherche | ANR-11-0001-02 PSL* | Romain Brette |
| Fonds Wetenschappelijk Onderzoek | G.0961.11 | Philip X Joris |

| Funder | Grant reference | Author |
|---|---|---|
| KU Leuven | Bijzonder Onderzoeksfonds OT/09/50 | Philip X Joris |
| Agence Nationale de la Recherche | ANR-10-LABX-0087 | Romain Brette |
| Fonds Wetenschappelijk Onderzoek | G.0A11.13 | Philip X Joris |
| KU Leuven | Bijzonder Onderzoeksfonds OT/14/118 | Philip X Joris |
| Fonds Wetenschappelijk Onderzoek | PhD fellowship | Tom P Franken |

The funders had no role in study design, data collection and interpretation, or the decision to submit the work for publication.

## Author contributions

VB, Recorded HRTFs, Conception and design, Acquisition of data, Analysis and interpretation of data, Drafting or revising the article; BF, Conception and design, Analysis and interpretation of data; TPF, Performed and analyzed the gerbil MSO recordings, Acquisition of data, Analysis and interpretation of data; SK, Performed the auditory nerve measurements and analysis, Acquisition of data, Analysis and interpretation of data; PXJ, Performed the IC recordings, Conception and design, Acquisition of data, Analysis and interpretation of data, Drafting or revising the article; RB, Conception and design, Analysis and interpretation of data, Drafting or revising the article

## Author ORCIDs

Philip X Joris, http://orcid.org/0000-0002-9759-5375

## Ethics

Animal experimentation: All procedures were approved by the institutional Animal Care Committee and were in accordance with the NIH Guide for the Care and Use of Laboratory Animals (P155/2008 to PX Joris (2009-2013)).

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
