## [Decision Letter]

Thank you for sending your work entitled “Frequency-dependent time differences between the ears are matched in neural tuning” for consideration at *eLife*. Your article has been favorably evaluated by Eve Marder (Senior editor), a Reviewing editor, and two reviewers, one of whom, Catherine Carr, has agreed to share her identity.

The Reviewing editor and the reviewers discussed their comments before we reached this decision, and the Reviewing editor has assembled the following comments to help you prepare a revised submission.

The manuscript is written well and figures are organized clearly. The authors make the case that environmental factors pose challenges to understand sound localization as simple mapping of azimuth through ITDs. While neither the HRTF or ITD results are new, the interpretation of them is, and calls into question the simplest interpretation of the currently dominant two-channel model. Thus, a major strength of this paper is the connection between the features seen in the electrophysiological data and the hypothesis that neurons are tuned to frequency-dependent ITDs as found in ecological environments. By analogy, incorporating statistics of natural stimuli have changed our understanding of early vision.

The concerns raised by the two reviewers are substantial and largely complementary to one another. Some of the concerns relate to clarifying what is new in the present study versus ground that has been covered in other publications noted by the reviewers.

1) A similar model and similar conclusions for the IC were previously published by Joris (PNAS, 2006). Please clarify whether the current analysis was based on the same data, and whether the main new finding is the tie to the ecologically appropriate HRTFs.

2) The concept of frequency mismatches and cochlear delays (stereausis), first proposed by [43] has been tested and rejected many times. To be more convincing, please thoroughly discuss results that show there are frequency differences in ipsi- and contralateral inputs into MSO.

3) With respect to testing the model, please discuss recently published data by [46] and [6], that cover some of the same ground as in this paper, with fewer neurons but with the more difficult recordings from MSO. Could some of your findings reflect processing in the MSO as opposed to in the IC?

4) The frequency-dependence of ITD at a particular azimuth (Figure 3) would be better presented with more detail. Fits to frequency vs ITD or IPD (Figure 3) for constant azimuth are presented for a bandwidth limited to a linear portion of the frequency vs. ITD plot. How were these bandwidths selected for frequencies lying near maxima or minima of these plots?

5) A few more examples of neural best phase (BP) vs frequency plots would convey the range of data. In Figure 1, and appearing in published studies, deviation from linearity is greatest at high frequency. In the acoustic data, the greatest deviation shown from Tollin and Koka is at low frequency. Are the neural data shown here not representative, or is the suggestion made from using taxidermy or spherical models that the Tollin and Koka data may not be so representative of a more typical environmental situation?

6) In Figure 4, distributions of CP and CD are provided, based upon the distribution of neural CFs. It would be useful to know the distribution of neural CFs in the sample and also which features of the ITD vs frequency at constant azimuth plots yield the varying values of CD and CP (at least positive vs negative values). That is, do particular features of these graphs depend upon the neural sample of CF, noting that the sample may not reflect the actual distribution of CF in the inferior colliculus?

7) Predictions of the inverse relationship between CD and CP are shown for acoustic measurements (Figure 4), but negative CDs seem not to be as prevalent as in the neural data (Figure 1). These data are collected for a spherical head model. If feasible, please consider exploring this idea using data measured by Tollin and Koka from living cats, to determine if a broader range of CPs and more negative CDs are achieved.

8) The general discussion would be more directed if you consider relative contributions of CF mismatch, acoustic environment and inhibition in generating BP vs frequency plots. Essentially, this discussion may help explain why there are relatively fewer non-physiological ITDs in Figure 6 than in Figure 1.

9) You apparently aim toward a test of your ideas whereby awake animals would listen to a range of frequencies emanating from a closely spaced set of azimuthal locations (Figure 3) during single unit recordings. Some suggestion of how, experimentally, to dissect acoustic and several neural mechanisms would help conclude this story.

---

## [Author Response]

*1) A similar model and similar conclusions for the IC were previously published by Joris (PNAS, 2006). Please clarify whether the current analysis was based on the same data, and whether the main new finding is the tie to the ecologically appropriate HRTFs*.

The [23] study was in fact quite different from this study. The focus in that study was on the problem of the dependence of best ITD on CF across populations of IC neurons. Best ITD was measured with delayed noise (not tones). Specifically, it was shown that 1) cochlear disparity can create significant delays; 2) a fixed distance mismatch along the cochleae can create an inverse correlation between best ITD and CF in binaural neurons.

The question we address is different. We ask what is the origin of the dependence of best ITD on *stimulus* frequency (not characteristic frequency), where this dependence is measured for each cell (not across cells). We show that this dependence measured in cells matches the acoustic dependence of ITD on frequency measured in HRTFs, which suggests that cells are tuned to acoustical properties of natural environments (rather than measuring fixed ITDs). Accordingly, our analysis of data has focused on characteristic phase (CP) and characteristic delay (CD), which are measures of the dependence of best ITD on stimulus frequency, and those were not addressed in [23].

In the same way, we showed here that the cochlear disparity model can produce significant frequency-dependence of best ITD (not CF-dependence) within a given binaural cell, as measured by CP. As discussed below, we also propose that a combination of cochlear mismatches and axonal delay mismatches can explain the variety of CP and CD observed in cells.

Finally, regarding the data, data shown in [23] are responses to noise, not to tones. In the experimental data we report, the frequency of the tonal stimulus is varied, in contrast with the previous study where the stimulus was a broadband noise that was identical for all fibers studied. This allowed us to compute pseudo-binaural curves for different frequencies, and in turn extract CP and CD measures for each cell.

We clarified these points in the revised Discussion.

*2) The concept of frequency mismatches and cochlear delays (stereausis), first proposed by*
[43]
*has been tested and rejected many times. To be more convincing, please thoroughly discuss results that show there are frequency differences in ipsi- and contralateral inputs into MSO*.

The stereausis model (first advanced by [40]) proposes that cochlear mismatches are used instead of axonal delay mismatches, to generate internal delays. We propose that cochlear mismatches are used in addition to axonal delay mismatches, to generate appropriate frequency-dependence of internal delays.

The stereausis model in its original form (not the one we propose) has been rejected in the barn owl nucleus laminaris, not in mammals. Our proposition addresses ITD tuning in mammals, but we can briefly comment on findings in the barn owl. It has been found in NL that CF mismatches between monaural inputs to a binaural cell are small but significant and asymmetrical (Peña et al., J Neurosci 2001; see Figure 5; see also [8]). However there was no correlation with ITD tuning, which contradicts the stereausis model (i.e. that the cochlear mismatch is the main determinant of best ITD). But this does not mean that CF mismatches did not contribute to ITD tuning at all. On the contrary, Peña et al. (J Neurosci 2001) report that the CF mismatches were large enough to create predicted delays of up to 50 µs, which is quite substantial for barn owls. Thus the available data in barn owls contradict the original stereausis model, but not ours. The CF mismatches that we show are necessary to obtain the required frequency-dependence of best ITD are small (40 Hz in Figure 6), unlike the large differences required in the original stereausis model.

In mammals, there is unfortunately little data on CF mismatches in the monaural inputs to MSO neurons. In cats, Yin and Chan (J Neurophy 1990) report (p 480) that BFs (not CFs) were generally similar for both ears: they differed by 0.2 octaves or less for 13 of the 18 cells. As we show on Figure 6, this order of magnitude is sufficient to produce the observed frequency-dependence of best ITD. In gerbils, Day and Semple (J Neurophy, 2011) also found frequency-dependent best ITDs in MSO neurons. They showed that those relationships could be explained using a periphery model by a combination of cochlear mismatches and axonal delays (i.e., similar to our proposition, not original stereausis), but unfortunately there was no direct physiological evidence in that study.

Thus, while the original stereausis model does not appear consistent with earlier data, our proposition is to strengthen the hypothesis, we show in the revised manuscript an analysis of in vivo intracellular recordings in gerbil MSO cells (Figure 7). It shows in 6 cells the monaural EPSP rate as a function of tone frequency for both sides. It appears that there can indeed be differences in frequency tuning between ipsi- and contralateral inputs. Additional experiments would be needed to relate those differences with the frequency-dependence of ITD tuning.

We added a paragraph in the Discussion to address these points.

*3) With respect to testing the model, please discuss recently published data by*
[46]
*and*
[6]*, that cover some of the same ground as in this paper, with fewer neurons but with the more difficult recordings from MSO. Could some of your findings reflect processing in the MSO as opposed to in the IC?*

We agree that the questions we address would be ideally addressed in the MSO. As pointed out, MSO recordings are more challenging and this study required a large number of cells and stimulus conditions to be able to make statistical comparisons with acoustics. The mechanism we propose indeed applies primarily to the MSO, and it is assumed that ITD tuning in IC is inherited from MSO. This is to be contrasted with the proposition that complex ITD tuning in IC results from the integration of MSO cells with different center frequencies (31).

In both papers mentioned above, it was shown that MSO neurons do have frequency-dependent ITD tuning, as we observed in our IC recordings. They also both suggest that those properties are primarily due to coincidences between excitatory inputs, as opposed to, for example, delays induced by inhibition. A recent intracellular in vivo study in MSO neurons shows that inhibition does not produce internal delays (Franken et al., Nat Neurosci 2015), contrary to previous claims (Brand et al., Nature 2002). This set of studies suggests indeed that our model and findings should primarily apply to the MSO. We added a paragraph in the Discussion. Finally, we added our own MSO data which address frequency tuning more directly than the MSO studies mentioned, and with intracellular recordings so a clean assessment could be made of the frequency tuning of the monaural inputs (point 3, above).

*4) The frequency-dependence of ITD at a particular azimuth (Figurr 3) would be better presented with more detail. Fits to frequency vs ITD or IPD (*Figure 3*) for constant azimuth are presented for a bandwidth limited to a linear portion of the frequency vs. ITD plot*. *How were these bandwidths selected for frequencies lying near maxima or minima of these plots?*

In the revised text, we inserted an additional figure (Figure 3—figure supplement 1) with more fits to frequency vs. IPD, which should clarify this point. The choice of bandwidth is explained in the Methods (roughly proportional to center frequency, reflecting cochlear filter bandwidths). Note that the linear regression is done not on the frequency-ITD plots, but on the frequency-IPD plots.

*5) A few more examples of neural best phase (BP) vs frequency plots would convey the range of data. In*
Figure 1*, and appearing in published studies, deviation from linearity is greatest at high frequency. In the acoustic data, the greatest deviation shown from Tollin and Koka is at low frequency. Are the neural data shown here not representative, or is the suggestion made from using taxidermy or spherical models that the Tollin and Koka data may not be so representative of a more typical environmental situation?*

As suggested, we inserted an additional figure in the revised text showing more examples (Figure 1—figure supplement 4). The high frequency points shown in Figure 1, which deviate from the linear regression, contributes very little to the response of the cell (see Figure 1, sync-rate) and therefore to the values of CP and CD. Figure 1—figure supplement 1 shows that linear regressions are in fact quite accurate for most cells. We address the acoustical question in response to the next remark below.

*6) In*
Figure 4*, distributions of CP and CD are provided, based upon the distribution of neural CFs. It would be useful to know the distribution of neural CFs in the sample and also which features of the ITD vs frequency at constant azimuth plots yield the varying values of CD and CP (at least positive vs negative values). That is, do particular features of these graphs depend upon the neural sample of CF*, *noting that the sample may not reflect the actual distribution of CF in the inferior colliculus?*

We now show the distribution of CF and BF in the cells (Figure 1—figure supplement 3). To address the question of dependence on CF, we performed the same analysis of acoustical CP and CD for low CF (<1 kHz) and high CF (>1 kHz), shown in Figure 4—figure supplement 2. The features are qualitatively similar.

*7) Predictions of the inverse relationship between CD and CP are shown for acoustic measurements (*Figure 4*), but negative CDs seem not to be as prevalent as in the neural data (*Figure 1*). These data are collected for a spherical head model. If feasible, please consider exploring this idea using data measured by Tollin and Koka from living cats, to determine if a broader range of CPs and more negative CDs are achieved*.

Figure 4 actually was based on Tollin and Koka’s data, not the spherical model. Indeed negative CDs are not as prevalent as in neural data. The proportion of negative CDs actually depends on the choice of distribution of azimuths, as shown in Figure 4—figure supplement 2 (we added percentages on the figure). In general, there tend to be more negative CDs when the azimuth is close to 0° or 180° (i.e. when ITDs are small). We discuss this point in more detail in the revised text, and added a figure showing the proportion of negative acoustical CDs as a function of azimuth (Figure 4—figure supplement 3).

*8) The general discussion would be more directed if you consider relative contributions of CF mismatch, acoustic environment and inhibition in generating BP vs frequency plots. Essentially, this discussion may help explain why there are relatively fewer non-physiological ITDs in*
Figure 6
*than in*
Figure 1.

We added a discussion of these different mechanisms. Note that CF mismatch and inhibition are two possible mechanisms; acoustic environment is not a mechanism, it is the target match for those two candidate mechanisms.

Note also that Figure 6 only includes CF mismatches, not axonal delay mismatches—this figure was meant to show that positive CPs can be obtained with CF mismatches, but we also postulate additional axonal delay mismatches.

*9) You apparently aim toward a test of your ideas whereby awake animals would listen to a range of frequencies emanating from a closely spaced set of azimuthal locations (*Figure 3*) during single unit recordings. Some suggestion of how, experimentally, to dissect acoustic and several neural mechanisms would help conclude this story*.

We added some text in the revised discussion at the two relevant places. Indeed there are two distinct aspects to be tested. One is that binaural neurons are tuned to complex binaural invariants rather than fixed ITDs, and testing this hypothesis would involve measuring spatial receptive fields with sounds that differ in spectrum of spectrotemporal envelope. Another possibility is to raise animals with manipulated acoustical cues (for example with an earplug), and observe how properties of ITD tuning are changed. The other aspect is that such tuning might be produced by a combination of CF and axonal delay mismatch. This is only a proposed mechanism, which we find plausible. It could perhaps be tested with binaural reverse correlation and/or retrograde tracing in MSO neurons, but of course this is quite challenging, especially if it has to be done in many cells.